# Inventory of Good Practices of Sustainable and Circular Phosphorus Management in the Visegrad Group (V4)

**Marzena Smol** [1,*], **Paulina Marcinek** [1,*], **Zuzana Šimková** [2], **Tomáš Bakalár** [2], **Milan Hemzal** [3], **Jiří Jaromír Klemeš** [3], **Yee Van Fan** [3], **Kinga Lorencz** [4], **Eugeniusz Koda** [5] and **Anna Podlasek** [5]

1 Mineral and Energy Economy Research Institute, Polish Academy of Sciences, J. Wybickiego 7A St., 31-261 Kraków, Poland
2 Faculty of Mining, Ecology, Process Control and Geotechnologies, Technical University of Košice, Park Komenského 19, 04 001 Košice, Slovakia
3 Sustainable Process Integration Laboratory–SPIL, NETME Centre, Faculty of Mechanical Engineering, Brno University of Technology–VUT Brno, Technická 2896/2, 616 69 Brno, Czech Republic
4 Bay Zoltán Nonprofit Ltd. for Applied Research, Kondorfa utca 1 St., H-1116 Budapest, Hungary
5 Institute of Civil Engineering, Warsaw University of Life Sciences (WULS-SGGW), Nowoursynowska 159 St., 02-776 Warsaw, Poland
* Correspondence: smol@meeri.pl (M.S.); marcinek@meeri.pl (P.M.)

**Abstract:** The most important raw material needed for food production is phosphorus (P), which cannot be replaced by other elements. P is listed as a Critical Raw Material (CRM) for the European Union (EU). It is an element essential for human nutrition and is used for fertiliser production. The key importance of P for human life is evidenced by the fact that if there were not enough P in fertilisers, we would only be able to feed 1/3 of the world's population. Unfortunately, in Visegrad Group (V4) countries, Poland, Slovakia, Czech Republic, and Hungary, there is a lack of mineral deposits of phosphate rock. Therefore, there is a strong need to cover the demand for the P by importing from countries of varying stability, both economic and political, such as Russia, China, or Morocco. It is risky; if the borders for deliveries of goods are closed, it may be impossible to meet the needs of P. On the other hand, V4 countries have large secondary P resources in P-rich waste, which are lost due to P is not recovered on an industrial scale. The paper presents the importance of P raw materials in V4, the revision of primary and secondary P sources that can be used in agricultural systems, as well as the structure of import and export of P raw materials in these countries. In addition, examples of good phosphorus recovery practices in the V4 countries are presented. They include a list of initiatives dedicated to the sustainable management of P resources, and examples of P recovery projects. Implementation of P recovery for internal P-rich waste in V4 could ensure the safety of food production in this region. Such and similar initiatives may contribute to faster independence of the V4 countries from the import of P raw materials.

**Keywords:** phosphorus; resources; critical raw materials; Visegrad Group; V4; sustainable management

## 1. Introduction

Phosphorus (P) is one of the most important nutrients needed to sustain life [1,2], with properties that cannot be replaced by any other element [3]. Moreover, P is non-renewable [4,5]. In 2014, the European Commission (EC) indicated phosphorus rock as one of the most important critical raw materials (CRMs) for the European economy [2], and P was placed on the CRMs list [6]. Then, in 2017 and in 2020, phosphate rock and P were also included in the updated lists of CRMs [7,8].

P has no metallic properties. P is classified as flu of non-metals [9]. P is the basic nutrient responsible for the growth of all living organisms with properties that cannot be substituted [10]. Moreover, P is the third major component (after potash and nitrogen) used in industrial fertilisers. P represents a crucial element of the food security system [11].

To reduce its dependence on external markets, the European Union (EU) has, in recent years, emphasised the need to look for alternative sources of P. One option is to recover this valuable raw material from selected waste streams. This approach is in line with the assumptions of the circular economy (CE), the EU economic model, which underlines that the transformation towards CE could bring significant economic and environmental benefits for the Member States, including V4. Activities in the field of recovery of raw materials, including P, are also part of the new EU strategy: the European Green Deal [12]. The initiatives for more sustainable management of P raw materials are part of the proposed Farm to Fork Strategy based on the principle of creating a fair, healthy, and environmentally friendly food system [13]. The transition to more sustainable food systems has already begun [14], but with current food production methods (based only on primary raw materials), feeding a population (in which there may be a threat of disruption in the delivery of these materials, e.g., as a result of closing borders, introducing restrictions on imports from selected countries) remains a challenge. Food production continues to pollute air, water, and soil, contribute to biodiversity loss and climate change, and consume vast amounts of natural resources, while a large proportion of the food produced is wasted. Poor quality food contributes to obesity and diseases such as cancer. The farm to fork strategy will address the use of fertilisers in agriculture [13], with a strong emphasis on the recovery of nutrients from waste [15].

The specific actions for the sustainable management of biogenic raw material resources (such as phosphorus) have been undertaken for many years in various regions of the EU and in individual Member States [3]. So far, however, no analyses have been conducted on the central region of the EU, which comprises the Visegrad Group consisting of four countries: Poland, the Czech Republic, Hungary, and Slovakia [16]. This region does not have phosphorus deposits, therefore, the demand for P raw materials (necessary for the production of fertilisers and food) is met only by imports. In the face of the threats of the 21st century, such as a pandemic, it seems reasonable to take action to ensure the safety of P raw materials in this region, as well as to intensify activities to subsidise P from available waste streams. Currently, the COVID pandemic is the greatest threat to modern economies. The V4 countries were the first in the EU to introduce restrictions to prevent the spread of the virus, which proves their great responsibility to residents [17]. Citizens stayed home, and the only thing they needed to survive was food. It showed that the greatest challenge is to ensure people's safety and access to food requires the provision of raw materials for food production.

The paper presents the importance of sustainable management of P raw materials in V4 countries. Primary and secondary P sources that can be used in agricultural systems are listed; import and export are presented. In addition, examples of good P recovery practices in the V4 countries are presented. The structure of the paper is as follows:

- clarify the importance of the recovery of P raw materials from waste in the context of V4 countries;
- overview of possible sources of P raw materials in V4 countries;
- overview of good examples of sustainable management of P raw materials in V4 countries;
- conclusions.

## 2. Materials and Methods

This section provides a description of the materials and methods that have been used in the study. The research framework is shown in Figure 1. There are four individual phases in this research. The first phase included a description of the case study region. The second phase included an overview of possible sources of P raw materials in V4 countries coming from primary and secondary sources. The third phase contained an overview of good examples of sustainable management of P raw materials in all V4 countries. The last phase covered conclusions from the study and recommendations for further research.

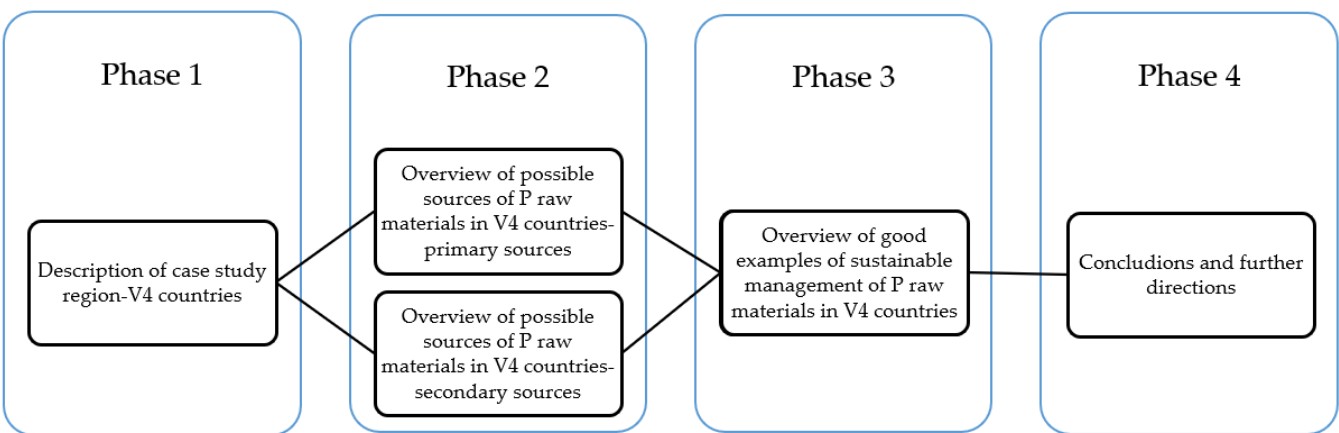

**Figure 1.** Scheme of the research framework.

In all phases of the work, a comprehensive analysis (desk research) of selected documents was used as a research method. The review covered numerous peer-reviewed scientific articles directly related to the subject of the flow of P raw materials in the V4 countries. The selection of the analysed literature was based on the following keywords: "phosphorus", "resources", "critical raw materials", "CRMs", "Visegrad Group", "V4", "sustainable management", "Czech Republic", "Hungary", "Poland", "Slovakia", "sewage sludge", "sewage sludge ash". The reviewed publications were searched on scientific platforms such as Elsevier Scopus and ScienceDirect, Multidisciplinary Digital Publishing Institute (MDPI), and Google Scholar. An important source of data was also a statistic published by Eurostat (the official statistic of the EU).

The initial results of this review were presented in the document "Portfolio of Phosphorus Friends in Europe", which was developed as part of the project "How to stay alive in V4? Phosphorus Friends Club builds V4's resilience (PhosV4)", financed by the Visegrad Fund (project no. 22110364). In this document, project partners contained information about the importance of P raw materials in securing the supply of food in V4 countries, the use of P raw materials in the food sector, and P raw materials flow in V4 countries. It is worth noticing that the identification of P recovery potential and good practices of P recovery in V4 countries is a research gap for which detailed data are not available at the moment. Therefore, it is an interesting research area that should be developed and studied. All results were discussed by project partners during the consortium meetings, and further directions for research were jointly designed.

## 3. Results

### 3.1. Case Study Region

The case study region in this paper is the V4 group, which contains the following four countries: the Czech Republic, Hungary, Poland, and Slovakia. The number of residents in this region it is above 63 M in 2022, with the higher number in Poland (37,654,247 people), followed by the Czech Republic (10,516,707 people), Hungary (9,689,010 people), and Slovakia (5,434,712 people) (Figure 2) [18].

Over the 11 years, the EU population has grown by around 6,887,000 citizens. Population growth occurred in Slovakia (by 43,000 in 2022 compared to 2011) and in the Czech Republic (population increase of 30,000 people). On the other hand, a decrease in the number of respondents took place in Poland (409,000 people less in 2022 compared to 2011) and in Hungary (a decrease in the population of 297,000 people in the analysed period).

The area of the EU countries currently covers 4,215,000 km$^2$, 5.2% of which is occupied by the V4 countries. Poland is the largest country belonging to the group of V4 countries, with an area of 312,700 km$^2$. Next is Hungary, with an area of 93,000 km$^2$, the Czech Republic at 78,900 km$^2$, and Slovakia, at 49,000 km$^2$, is the country in the region with the smallest area [19].

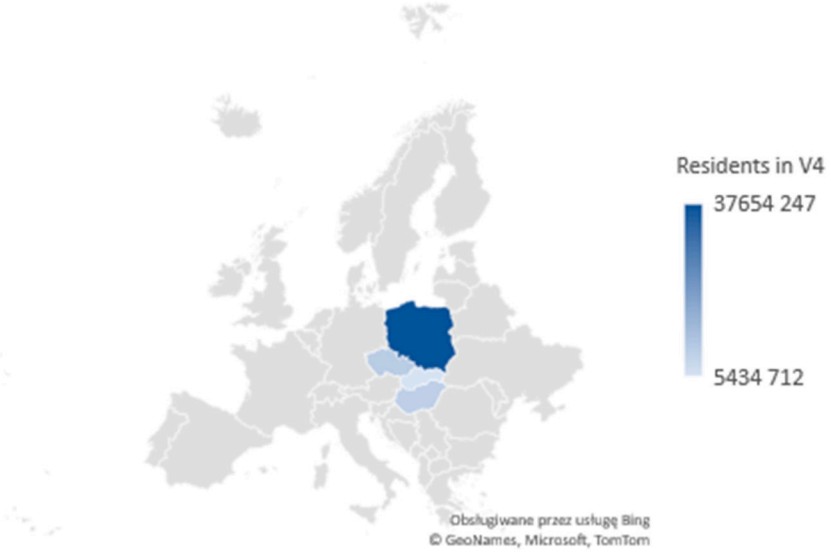

**Figure 2.** Residents in V4 [18].

The changes in population over 11 years (2011–2022) in the V4 countries is presented in Table 1.

**Table 1.** Population in V4 (in thousand) (data from [18]).

| Year | 2011 | 2012 | 2013 | 2014 | 2015 | 2016 | 2017 | 2018 | 2019 | 2020 | 2021 | 2022 |
|---|---|---|---|---|---|---|---|---|---|---|---|---|
| EU-27 countries | 439,942 | 440,553 | 441,258 | 442,884 | 443,667 | 444,803 | 445,534 | 446,209 | 446,559 | 447,485 | 447,001 | 446,829 |
| Hungary | 9986 | 9932 | 9909 | 9877 | 9856 | 9830 | 9798 | 9778 | 9773 | 9770 | 9731 | 9689 |
| Poland | 38,063 | 38,064 | 38,063 | 38,018 | 38,006 | 37,967 | 37,973 | 37,977 | 37,973 | 37,958 | 37,840 | 37,654 |
| Slovakia | 5392 | 5404 | 5411 | 5416 | 5421 | 5426 | 5435 | 5443 | 5450 | 5458 | 5460 | 5435 |
| Czech Republic | 10,487 | 10,505 | 10,516 | 10,512 | 10,538 | 10,554 | 10,579 | 10,610 | 10,650 | 10,694 | 10,495 | 10,517 |

### 3.2. Primary and Secondary P Raw Materials in V4 Countries

In the EU, P resources are limited, which means that most of the P in the EU is imported. The EU imports around 6 million mg of natural phosphate annually and around 1.2 million mg of P fertilisers from Russia, Morocco, and Tunisia [20]. EC presents 100% as the reliance percentage on P imports and 84% as the reliance percentage on phosphate rock [8]. The structure of global producers and main EU sourcing countries of phosphate rock is presented in Figure 3.

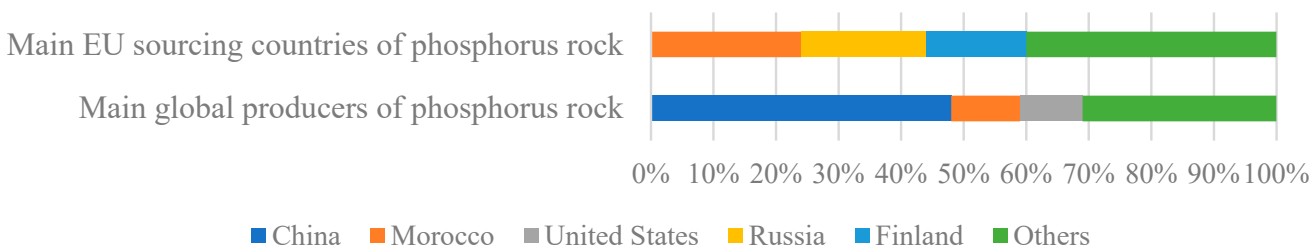

**Figure 3.** Structure of global producers and main EU sourcing countries of phosphate rock, data from [21].

Based on the available data, there are no P deposits mined in V4 countries at present. In Poland, there are phosphorite deposits that were mined in the past. P occurs at the northeastern margin of the Holy Cross Mts. (vicinities of Radom-Iłża-Annopol-Gościeradów-Modliborzyce) in the form of calcium phosphate-rich nodules in the various types of sediment [22]. The exploitation of phosphate phosphorites began in the country between

the First and the Second World Wars. Currently, due to economic aspects, no deposits are exploited. The last exploited phosphorite deposit, located in Chałupki, was closed in 1961, and 10 years later, the same was also done in Annopol [23]. The limit values of the parameters that describe the phosphorites deposit in Poland define that [22]:

- the maximum depth of deposits documentation is 400 m below the surface,
- the minimum $P_2O_5$ content in calcium phosphate-rich nodules is 15%,
- the minimum affluence of calcium phosphate-rich nodules is 1800 kg/m$^2$.

Qualitative parameters of the main phosphorites occurrences in Poland are presented in Table 2.

**Table 2.** Quality parameters of documented phosphate deposits (data from [23]).

| Deposit | Calcium Phosphate-Rich Nodules (mm) | $P_2O_5$ Content in Calcium Phosphate-Rich Nodules (%) | Affluence of Calcium Phosphate-Rich Nodules (kg/m$^2$) | Affluence Versus Actual Limiting Parameters (%) |
|---|---|---|---|---|
| Annopol | >10 | 13.5 | 568 | 32 |
| Burzenin | >2 | 18.1 | 385 | 21 |
| Chałupki | >10 | 14.9 | 354 | 21 |
| Gościeradów | >2 | 15.2 | 496 | 28 |
| Iłża-Krzyżanowice | >2 | 18.6 | 791 | 44 |
| Iłża—Chwałowice | >2 | 22.3 | 891 | 50 |
| Iłża—Łęczany | >2 | 18.6 | 654 | 36 |
| Iłża—Walentynów | >2 | 19.9 | 470 | 26 |
| Radom—Dąbrówka Warszawska | >2 | 16.5 | upper series: 317 lower series: 460 | upper series: 18 lower series: 26 |
| Radom—Krogulcza | >2 | 19.1 | upper series: 218 lower series: 504 | upper series: 12 lower series: 28 |
| Radom—Wolanów | >2 | 15.4 | upper series: 170 lower series: 447 | upper series: 9 lower series: 25 |

The index describing the abundance largely deviates from the boundary values of the parameters that define the deposit. Deposits are flooded, which results in their potential exploitation. In addition, railway lines and high-voltage lines, roads or buildings were built in their areas through significant parts of the deposits. In extreme cases, it may cause the resources available for exploitations reduction as much as 50–80%.

All deposits from which phosphate rock was obtained in Poland were removed from the national resource balance in 2006. Currently, the domestic demand for phosphate rock raw materials is fully covered by imports, e.g., from Morocco, Algeria, and Egypt, where the availability of the described raw materials is much greater and more economical [22]. The phosphate rock import quantity to Poland during the last 18 years is presented in Figure 4. The largest amount of the P was imported in 2004, and the lowest amount in 2009, which is directly related to the global economic crisis that occurred in 2008.

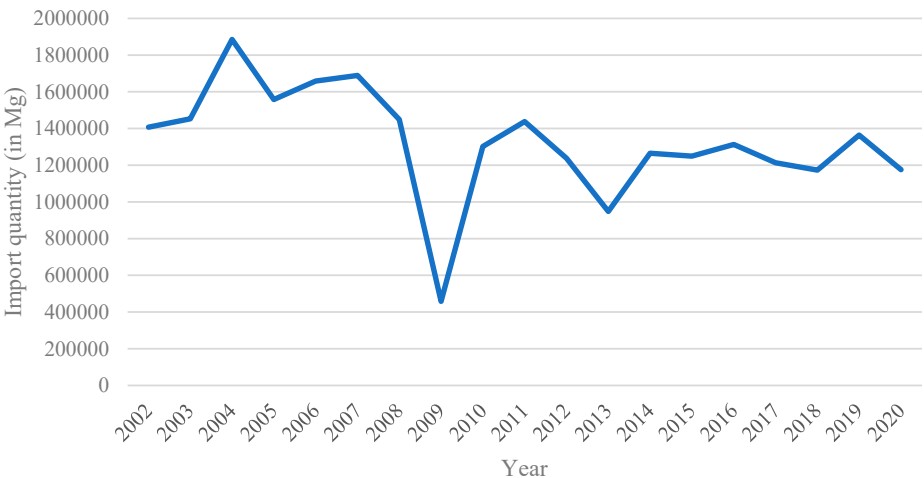

**Figure 4.** Phosphate rock import quantity to Poland, data from [24].

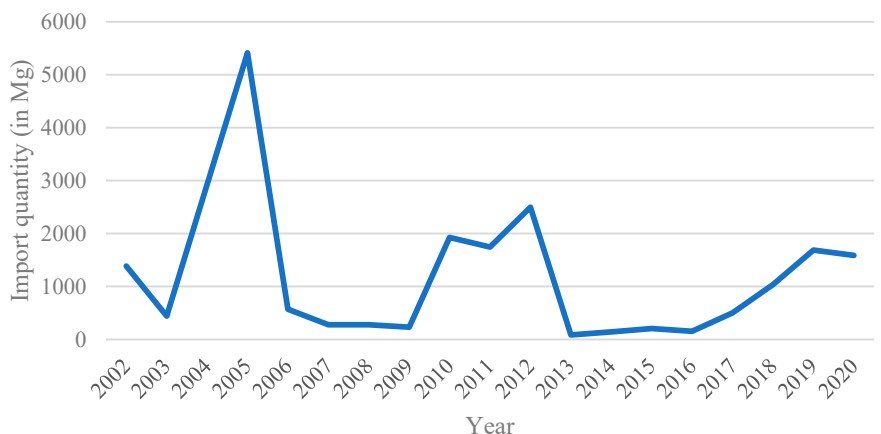

**Figure 5.** Phosphate rock import quantity to Slovakia, data from [24].

In the Czech Republic, there are no P deposits. The domestic demand for phosphate rock raw materials is fully covered by imports. The phosphate rock import quantity to the Czech Republic during the last 18 years is presented in Figure 6. The largest amount of the P was imported in 2008, and the lowest amount was in 2012.

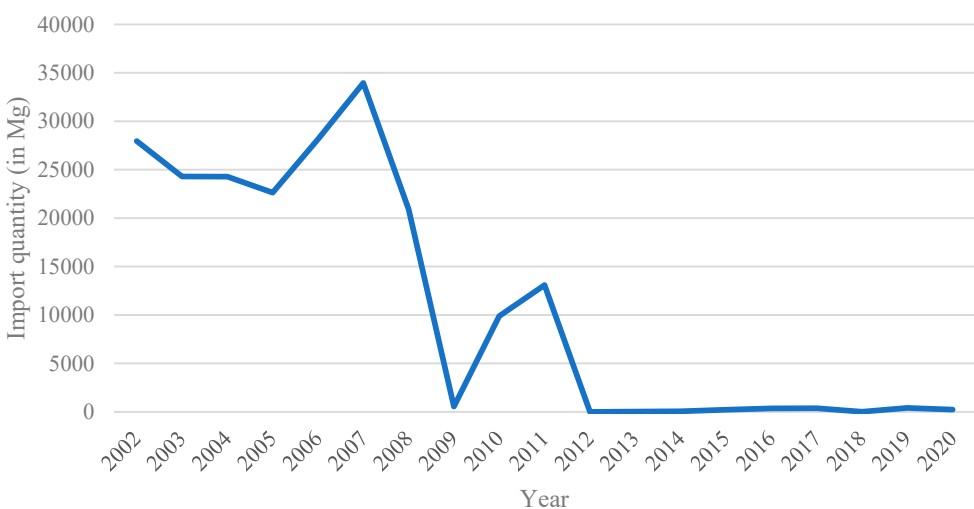

**Figure 6.** Quantity of imported phosphate rocks in the Czech Republic, data from [24].

In Slovakia, there is one deposit of phosphate; however, it also is not mined at the moment. The demand for phosphate raw materials is fully covered by imports. The phosphate rock import quantity to Slovakia during the last 18 years is presented in Figure 5. The highest amount of P raw materials was imported to Slovakia from Italy (68%) and the Czech Republic (31%). There is also limited import from Germany (0.2%), the United Kingdom, Belgium, Japan, and other countries (<0.1%) [25,26].

In Hungary, there are five sedimentary phosphate deposits, but there is no information available on what phosphorus contents there are and whether they will ever be used. The phosphate rock import quantity to Hungary during the available 10 years is presented in Figure 7. The largest amount of the P was imported in 2020, and the lowest amount was in 2014.

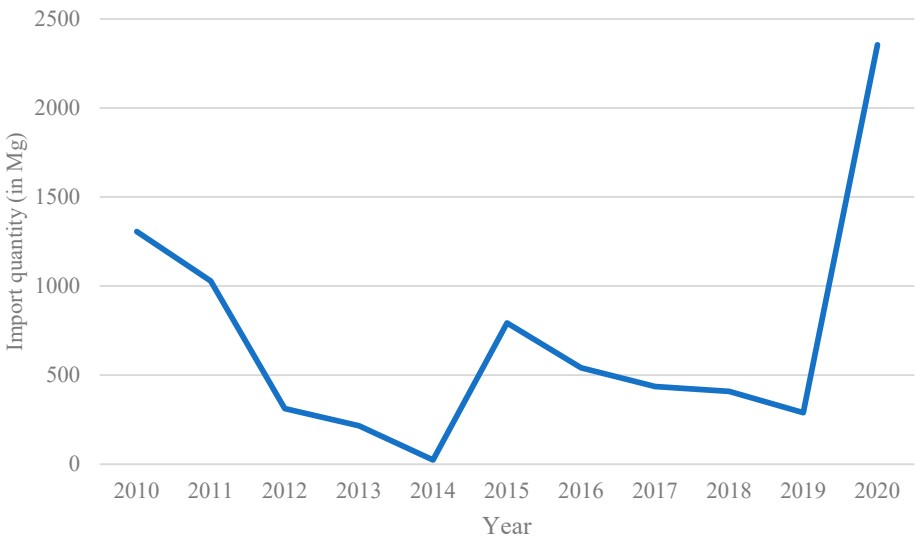

**Figure 7.** Quantity of imported phosphate rocks in Hungary, data from [24].

The vital importance of P and its growing deficiency influenced the dynamic development of science in the area of P recovery from different waste materials [27]. In V4 countries, there is high potential for the recovery of P from secondary sources such as:

- wastewater and sewage sludge (municipal and industrial) [28],
- sewage sludge ash [29],
- pig slurry [30],
- meat and bone meal [31],
- industrial waste [32],
- biomass [33].

The P contents in selected waste streams are shown in Table 3, which includes values of P concentration in sewage sludge, ashes from sewage sludge, animal manure, and compost from plant waste.

**Table 3.** P concentration in presented types of waste [mg/kg].

| Types of Waste | P Concentration [mg/kg] | | | | References |
|---|---|---|---|---|---|
| Sewage sludge | 13,200 | 123,000 | 26,100 | 65,400 | [34–37] |
| Sewage sludge ash | 46,200 | 60,697 | 127,351 | 112,425 | [37–40] |
| Animal manure | 21,400 | 30,600 | 32,700 | 29,500 | [41–44] |
| Compost from plant waste | 40,900 | 89,000 | 83,000 | 78,000 | [35,45–47] |

Currently, household waste containing large amounts of P (mainly sewage sludge) could cover around 20–30% of the demand for phosphate fertilisers in the EU when recycled. However, this investment potential is still largely untapped in European countries [48],

despite the fact that such an approach is in line with the assumptions of CE, in which waste generated should be treated as secondary raw materials. In V4 countries, municipal wastewater treatment plants (WWTPs) have the greatest potential for P recovery because P can theoretically be recovered at every stage of the treatment process, i.e., from sewage and leachates in the liquid phase, from dehydrated sewage sludge, from the solid phase of ashes after thermal transformation of municipal sewage sludge. In the successive stages of wastewater treatment and sewage sludge treatment, a smaller volume of the substrate used for P recovery is observed, while at the same time P concentration per unit volume is increasing [49]. The efficiency of P recovery from different substrates at WWTPs is equal to [50]:

- 45–55% for wastewater-outflow from the treatment plant,
- 45–50% for sedimentary liquid – leachate,
- 50–60% for dehydrated sewage sludge,
- >90% for sewage sludge ash.

The major part of P in substrates in WWTPs is transferred to sludges (up to 90%) [40]. Therefore, sludge and sludge ash are the most promising P-rich residues. Sewage sludge production and disposal from urban wastewater in V4 countries are presented in Table 4. There is an increasing amount of sewage sludge generated in V4 countries. A higher amount of SS is observed in Poland, which corresponds to the highest population in this country, followed by Hungary, the Czech Republic, and Slovakia. In total, 107,845 thousand mg of SS was produced in V4 countries in 2019 [51].

**Table 4.** Sewage sludge production and disposal from urban wastewater in V4 countries in 2009–2019, in thousand mg (data from [51]).

| Country | 2009 | 2010 | 2011 | 2012 | 2013 | 2014 | 2015 | 2016 | 2017 | 2018 | 2019 |
|---|---|---|---|---|---|---|---|---|---|---|---|
| | | | | | | Thousand mg | | | | | |
| Czech Republic | 20,720 | 19,630 | 21,790 | 26,330 | 26,010 | 23,859 | 21,024 | 20,671 | 22,327 | 22,822 | 22,109 |
| Hungary | 14,930 | 17,034 | 16,833 | 16,060 | 17,047 | 16,312 | 17,770 | 21,796 | 26,684 | 23,366 | 22,789 |
| Poland | 56,330 | 52,670 | 51,920 | 53,330 | 54,030 | 55,600 | 56,800 | 56,833 | 58,445 | 58,307 | 57,464 |
| Slovakia | 5858 | 5476 | 5872 | 5871 | 5743 | 5688 | 5624 | 5305 | 5452 | 5593 | 5483 |
| Total V4 | 97,838 | 94,810 | 96,415 | 101,591 | 102,830 | 101,459 | 101,218 | 104,605 | 112,908 | 110,088 | 107,845 |

There are several technologies for P recovery from SS, including P extraction by wet chemical methods under acid and alkali conditions [50]. However, to date, there is no reported industrial plant that is recovering P raw materials from SS in V4 countries. Therefore, further initiatives (as economic, environmental, law, or social) that support P recovery technologies implementation from SS should be developed.

The highest efficiency of P recovery was reported for the ashes generated in the process of thermal treatment of sewage sludge (>90%). In the V4 group, only Poland is equipped with municipal sewage sludge incineration plants (so-called mono-incineration plants). The detailed inventory of SSA generated in Poland was reported in [52]. The current capacity of 11 mono-incineration plants is equal to 160,300 mg d.w. of SS per year. The highest amount of ashes is generated in Warsaw and Cracow. The most important player is a mono-incineration plant in Warsaw (the capital of Poland) that produced >10,000 mg d.w. in 2018 (38% of total SSA generated in Poland). There are also significant amounts of ashes in Cracow (18% of the total in Poland), Łódź (14%), Gdańsk (14%), and Gdynia (12%). The rest of the installations produced less than 10% (6%—Gdynia, 5%—Szczecin, 3%—Kielce, 2%—Bydgoszcz, 1%—Olsztyn). In total, in 2018, 24,510 of fly ash and 24,510 mg of bottom slag and ash were produced. They potentially can be used in phosphate fertiliser production; however, for economic reasons, there is no industrial processing and production of P fertilisers from this waste stream. To protect the utility value of ashes, they have to be stored selectively and then directed to P recovery. This supports the possibility of turning waste into a resource if certain conditions are met. Despite the high content of P raw materials in the ashes, it is usually present in chemically bound forms, which makes its

availability to plants difficult. In addition, the ashes may contain significant amounts of impurities, including heavy metals, which limits the possibility of their use in the production of fertilisers without prior treatment. In order to increase the bioavailability of phosphorus and reduce the content of heavy metals, this waste should be subjected to chemical and thermochemical processing. The most promising methods of phosphorus recovery from ashes are chemical methods (using phosphorus extraction-wet methods) and thermochemical methods (separation of P fraction at high temperatures 1000–2000 °C and conversion of phosphorus into forms available for plants) [52].

### 3.3. Good Practices of P Recovery in V4 Countries

This section includes an inventory of selected examples of good practices of sustainable P management in V4 countries. They include a list of innovative solutions enabling the recovery of P raw materials from different waste streams.

### 3.3.1. Good Practices of P Recovery in Poland

In Poland, many activities are undertaken that are dedicated to sustainable and circular management of P. There are several projects in the country, the aim of which is, inter alia, P recovery. Moreover, many companies take measures to support the acquisition of P from secondary sources. Table 5 shows examples of good P management practices in Poland.

**Table 5.** Examples of good practices P management in Poland.

| Company/Project Name | Description of Good Practices | References |
|---|---|---|
| Jarocin Waterworks Company | The Jarocin Waterworks Company has signed a contract for carrying out an investment under the project 'Modernisation and Extension of WWTP Jarocin'. The project includes the implementation of five tasks (with a total value of 60 M EUR), supported by co-financing from the EU. The largest investment in the project is the construction of a station for the recovery of raw materials, such as nitrogen, P, and biogas, at the sewage treatment plant in Cielcza. This would allow it to recover between 100 and 200 kg of fertiliser per year. The water and wastewater management project implemented in Jarocin was recognised with a prestigious award at the international Wex Global 2018 conference, which took place in Lisbon. In the years that follow, the introduction on the market of technologies for the recovery of P will be planned, in particular in the wastewater sector. | [53] |
| Azoty Group "Fosfory" | Azoty Group "Fosfory" Sp. z o. o. are one of the leaders in the fertiliser and chemical industry in Europe. The highest quality of products and complete customer satisfaction are their priority. By producing agricultural fertilisers, they strive to maximise the benefits of buyers and maintain all environmental protection requirements. Group is a producer of mineral fertilisers that are widely used in agriculture, vegetable cultivation, and horticulture. Their offers also included chemical products. The Azoty Group "Fosfory" taking advantage of the location within the Gdansk port and access to the Chemikow Wharf in use, it imports some raw materials for the production of fertilisers by sea. With its experience in maritime trading, the Azoty Group "Fosfory" also conducts a wide range of services and reloading as well as sea freight of loose and liquid bulk goods in export and import. | [54] |

**Table 5.** *Cont.*

| Company/Project Name | Description of Good Practices | References |
|---|---|---|
| Sewage Treatment Plant—Tarnowskie Wodociagi | Tarnowskie Wodociągi Sp. z o.o. provides services in the field of collection and treatment of municipal wastewater. In 2007, on the premises of Sewage Treatment Plant-Tarnowskie Wodociagi Sp. z o.o. the construction of a sewage sludge drying plant was started. This investment was completed in 2008. The construction of the dryer is another step towards even more complete use of sewage sludge and reducing its mass four times. Ultimately, it is planned to utilise sewage sludge along with recovery of P compounds from the ashes. | [55] |
| Project "Sustainable management of phosphorus in the Baltic region (InPhos)" | Project "Sustainable management of phosphorus in the Baltic region" (InPhos) received funding from the European Institute of Innovation and Technology (EIT)—a body of the EU, under the Horizon 2020 program. The main objective of the InPhos project was to develop a strategy for sustainable P management (including identification of the P recovery potential) in the Baltic Sea Region by a working group of experts from the Baltic countries—Poland, Germany, Sweden, Finland, Latvia, Lithuania, Estonia, and Italy. | [56] |
| Project "Market ready technologies for P-recovery from municipal wastewater (PhosForce)." | The main objective of the "Market ready technologies for P-recovery from municipal wastewater" (PhosForce) project is to develop innovative technology for the recovery of P from wastewater. The Struvia® technology has been used to recover P in the form of struvite crystals from wastewater generated in municipal waste disposal facilities. | [57] |
| Project "Towards Circular Economy in wastewater sector: Knowledge transfer and identification of the recovery potential for Phosphorus in Poland (CEPhosPOL)." | The main goal of the "Towards Circular Economy in wastewater sector: Knowledge transfer and identification of the recovery potential for Phosphorus in Poland" (CEPhosPOL) project was to conduct research works focused on the identification of the recovery potential for P in Poland and the development of the sustainable model of the P management, based on the circular economy assumptions. The project was implemented under the Mieczysław Bekker programme for young researchers, financed by the National Academic Exchange Agency (NAWA). | [58] |
| Project "Polish Fertilisers form Ash (PolFerAsh)" | The main goal of the Polish project "Polish Fertilisers form Ash" (PolFerAsh) was to develop an environmentally-friendly technology for sewage sludge ash utilisation as a source of fertilisers and construction materials. The project has been conducted in the Cracow University of Technology and Mineral and Energy Economy Research Institute of the Polish Academy of Sciences in Poland and has received the founding from the National Centre for Research and Development. | [59] |

### 3.3.2. Good Practices of P Recovery in Slovakia

In Slovakia, the company that produces fertilisers is Duslo, a.s. [60], which has become a fertiliser producer on a European and on a global scale. In addition, the country has undertaken actions aimed at sustainable P management, for example, through the project "Drinking water supply, sewerage and wastewater treatment" [61] or the Slovak Grant Agency for Science (Grant No. 1/0563/15) [62]. These activities are presented in Table 6.

### 3.3.3. Good Practices of P Recovery in the Czech Republic

In the Czech Republic, there are projects, institutions and organisations that support the sustainable development of P management. It is worth noticing that there is a national platform dedicated to P management–Czech Phosphorus Platform [63], which is an organisation that allows its members to act in the field of, inter alia, reducing dependence on imports and recycling of P from waste, from crop and livestock production in agriculture, from industry and municipal sewage. The activities of the Czech community in the field of sustainable P management are presented in Table 7.

### 3.3.4. Good Practices of P Recovery in HUNGARY

The leading Hungarian fertiliser partner network is called Genezis. This partner network includes five large companies, the activities and best practices of which are presented in Table 8.

**Table 6.** Examples of good practices P management in Slovakia.

| Company/Project Name | Description of Good Practices | References |
| --- | --- | --- |
| Duslo, a.s-company dealing with fertilisers | The biggest company dealing with fertilisers is Duslo, a.s., a member of the AGROFERT Group. It is one of the most significant companies in the chemical industry in Slovakia. It has developed into a manufacturer of fertilisers of European significance and a global supplier of rubber chemicals. It is a producer of polyvinyl acetate and polyacrylic glues and dispersions that it supplies to the global market. The company s product portfolio includes: industrial fertilisers, rubber chemicals, dispersions and glues, products of magnesium chemistry, and special products. | [60] |
| Project "Drinking water supply, sewerage and wastewater treatment." | The "Drinking water supply, sewerage and wastewater treatment" project contributed to reducing pollution and improving wastewater collection. The project also brought drinking water to people struggling to find regular or reliable supplies. As part of the project, the existing facilities were modernised and a new central pumping station was constructed. Improvements to existing facilities included making it easier to remove nitrogen and P from the water. These actions resulted in a radical increase in the capacity and efficiency of the existing wastewater plant. | [61] |
| Slovak Grant Agency for Science (Grant No. 1/0563/15) | The research project was carried out as planned research projects of the Department of Applied Ecology, Sumy State University, connected with subjects "Reduction of technogenic loading on the environment of enterprises of chemical, machine-building industry and heat and power engineering" according to the scientific and technical program of the Ministry of Education and Science of Ukraine (state registration No 0116U006606). The project focused, inter alia, on the biochemical treatment of sewage sludge and phosphogypsum under conditions reducing sulphates with the release of P. A schematic model of the dephosphatation process under the conditions of anaerobic stabilisation of sewage sludge and phosphogypsum was developed. | [62] |

**Table 7.** Examples of good practices P management in the Czech Republic.

| Company/Project Name | Description of Good Practices | References |
|---|---|---|
| Fosfa, a.s. | Fosfa, a.s. is an innovative Life Science company, the largest processor of yellow P in Europe and a successful exporter. After the successful resumption of phosphoric acid production, the company decided to invest in the production of special applications based on P and detergents. At present, Fosfa products are for food and alcohol industrial applications. The production scope of the company consists of product groups: sodium phosphates, potassium phosphates, ammonium phosphates, and thermal phosphoric acid. During production, the company keeps principles of sustainable development and footprint reduction strongly. | [64] |
| Lovochemie, a.s. | Lovochemie, a.s., is the largest producer of fertilisers in the Czech Republic. Its production program has significantly contributed to the development of Czech agriculture. The company decided to invest in the production of special applications based on P and detergents. Currently, the company produces NPK fertilisers. Lovochemie is trying to find long-term sustainable sources of P to replace current raw materials in future. | [65] |
| Czech Phosphorus Platform | Czech Phosphorus Platform is an organisation that brings together private companies, government agencies, academic institutions, and individuals. The organisation creates conditions for various activities of members in the area of recycling, circular economy, waste management, sustainable agriculture, and water management to reduce dependence on imports and to recycle P from waste, from crop and animal production in agriculture, from industrial and municipal wastewater. | [63] |
| Cleaning of the Brno lake from excess P | The Brno lake is the largest reservoir in Brno, measures 10 km in length, and the flooded area is 259 ha. The main problem of Brno lake for a long time was green cyanobacteria, which polluted the entire water area and made recreation impossible. Water purification and treatment in the lake began in 2007. The project on how to stop and improve the gradual deterioration of water quality at Brno lake, especially from flushing water, is called "Implementation of Measures at the Brno Valley Reservoir", which aims to reduce the effects of excessive eutrophication on water. The aeration system, in combination with ferric sulphate dosing, ensures the precipitation of P, which sinks to the bottom and becomes (so far) its harmless part. Results show an improvement in water quality in the lake. Applied systems are used to precipitate P, which is the main food for bacteria and most often enters the water with rainwater from fields where farmers use it as fertiliser. These measures have proved very successful over the years and therefore continue during the next stage of the project, "Implementation of measures at the Brno Valley Reservoir, IVth stage 2023–2027 ". The project is managed and implemented by the Moravia River Basin District. Its staff monitors the state of the water, monitors the health of aquatic animals, and generally finds out how the dam is doing thanks to continuous care. The next significant necessary steps are the removal of precipitated P from the bottom of the Brno Lake and the recovery of P by recycling. | [66] |

**Table 8.** Examples of good practices P management in Hungary.

| Company/Project Name | Description of Good Practices | References |
| --- | --- | --- |
| Bige Holding Ltd. | Bige Holding Ltd. privatised the Tiszamenti Vegyiművek Rt. in 1997. Following the investment in 2004, Bige Holding Ltd. produces compacted NPK products from the Genesis fertiliser range, as well as sulphuric acid, phosphoric acid, and cryolite. The plant, which operates with compacting technology, produces reliably high-quality NPK and PK fertilisers without chemical reaction and drying process, with both meso- and micro-nutrient content in the quality required by the customer. The particle size and strength of the fertilisers are produced to meet today's modern European quality standards. The environmental impact of the new technology is minimal. | [67] |
| Nitrogénművek Zrt. | Nitrogénművek Zrt. in Pétfürdő is the one Hungarian nitrogen fertiliser company with ammonia and fertiliser production capacities. The range includes nitrogen fertilisers, complex NPK fertilisers, foliar and nutrient fertilisers. Chemical products and industrial gases generated during the fertiliser production process are also sold. The main task of the company is to meet the long-term demand for fertilisers in Hungarian agriculture. The current market share of Nitrogénművek Zrt. in the domestic fertiliser market is about 60%. | [68] |
| Péti Nitrokomplex Ltd. | Péti Nitrokomplex Ltd. is owned by Nitrogénművek Zrt, which was founded in 1991 by the self-establishment of the research and development part of the plant. The main goal of the company is to meet the needs of its customers and to adapt to the principles of environmentally friendly, integrated crop production, i.e., the rational supply of nutrients according to the area and the needs of the plan. | [69] |
| Nádudvar Agrochemical Ltd. | Nádudvar Agrochemical Ltd. operates a world-class state-of-the-art liquid fertiliser service system. The primary objective of the agricultural plant is economic production, which has necessitated the application of state-of-the-art methods in the crop production sector. To achieve this goal, created one of the most advanced liquid fertiliser plants of the time. The company's services include consultancy, transport, the setting up of transit depots, the provision of a group of application machines and the development of using technology. | [70] |
| Nzrt-Trade Ltd. | Nzrt-Trade Ltd. is a fertiliser supplier in the eastern part of Hungarian agriculture; as a member of the Bige Holding Group, it has a significant R&D activity in the production of fertilisers. The company has links with several research institutes and universities, which carry out the crop certification of its products and the basic research work necessary for their development. | [71] |

## 4. Conclusions

Sustainable management of mineral resources is an important element of functioning of European countries, including the V4 countries. P is one of the most important elements that belongs to the group of CRMs and is an essential element of human nutrition. Moreover, P cannot be replaced by another element. What is more, there is a problem with limiting P resources and planetary boundary for phosphorus is clearly exceeded. The V4 countries do mine P raw materials, and they satisfy the demand with imports. It is possible to replace the current imports of the V4 countries with raw materials from secondary sources, such as:

- industrial wastewater,
- biomass,
- industrial waste,
- others.

Currently, the identification of P recovery potential and good practices of P recovery in V4 countries is a research gap for which detailed data are not available. Moreover, despite the current economic crisis, fertilizers from primary sources are still cheaper than fertilizers from secondary sources. For this reason, in the V4 countries, the topic of good practices in the context of obtaining alternative fertilizers from secondary sources is not popular. Nevertheless, V4 countries have taken steps to broaden the knowledge of P raw materials in society. Initiatives that disseminate information on P raw materials include organisations promoting innovative solutions for the extraction and sustainable management of P or projects related to P raw materials in which countries participate. Such projects include "How to stay alive in V4? Phosphorus Friends Club builds V4's resilience", whose main goal is to increase the knowledge and awareness of the importance of P raw materials for food production in the V4 countries. The project also aims to develop a strategy for the sustainable management of P, which will contribute to ensuring a sufficient amount of P for food production. It also includes various awareness-raising events such as a workshop and a follow-up conference. Project products as a P management roadmap in V4 countries will accelerate the implementation of P recovery and ensure the safety of food production during and after the COVID pandemic. Such and similar initiatives may contribute to faster independence of the V4 countries from the import of P raw materials.

**Author Contributions:** Conceptualisation, M.S., P.M. and J.J.K.; methodology, M.S. and P.M.; investigation, M.S.; P.M.; Z.Š.; T.B.; M.H.; J.J.K.; Y.V.F.; K.L.; E.K. and A.P.; resources, M.S.; P.M.; Z.Š.; T.B.; M.H.; J.J.K.; Y.V.F.; K.L.; E.K. and A.P.; writing—original draft preparation, M.S. and P.M.; writing—review and editing, M.S. and P.M.; visualisation, M.S. and P.M.; supervision, M.S. and J.J.K. All authors have read and agreed to the published version of the manuscript.

**Funding:** This research received no external funding.

**Data Availability Statement:** The datasets generated during and/or analysed during the current study are available from the corresponding author on reasonable request.

**Acknowledgments:** The study was developed under the project: PhosV4—"How to stay alive in V4? Phosphorus Friends Club builds V4's resilience", no. 22110364 (2021–2023), which is financed by Visegrad Fund.

**Conflicts of Interest:** The authors declare no conflict of interest.

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
