# Peer review of "Inventory of Good Practices of Sustainable and Circular Phosphorus Management in the Visegrad Group (V4)"

_resources, doi:10.3390/resources12010002_

Round 1
Reviewer 1 Report
The article "Inventory of good practices of sustainable and circular phosphorus management in the Visegrad Group (V4)" presents the activities in four EU member states in Eastern and Central Europe to mamage phosphorus as a critical resource. The manuscript has signifcant lengths . For example, it does not contribute to the topic to present the position of phosphorus in the periodic table and explaining the four allotropes of elemental phosphorus.It would be more interesting to inform about the chemical identity of the natural deposits and the fertilizers derived from them and the extent to which theey are bioavailable. Also, the reference should not be missing that the planetary boundary for phosphorus is clearly exceeded and the goal is to reduce the amount of emitted phosphate and to fertilze only accourding to the needs of the crops.
The goal to recycle phosphorus and to make accessible secondary phosphorus sources such as sewage sludge, biowaste and other wastes is clearly stated by the authors. Therefore, section 3.3 "Good practices of P recovery in V4 countries" is the key information. However, it is surprising that many manufacturers are listed without mentioning their P recovery activities. In the end, only three projects are presented that focus on phosphorus recovery: PhosForce and PolFerAsh in Poland and the Slovak Grant Agency for Science.
Apparently, recovery of phosphorus from wastewater and sewage sludge does not take place in the V4 countries, even if mono-incineration is practised (page 10). An explanation of the economic reasons and a comparison with other EU countries should be added.
The sustainable management of phosphorus also includes the development of agricultural practices that reduce the input of phosphorus into the environment and thus prevent damage such as the hypertrophication of Brno lake.

Author Response
Dear Reviewer #1,
I am enclosing here with a manuscript entitled “Inventory of good practices of sustainable and circular phosphorus management in the Visegrad Group (V4)” submitted to “Resources” for possible evaluation after your review. With the submission of this manuscript I would like to undertake that the above mentioned manuscript has been corrected according to Your and other Reviewers comments.
Reviewer comments: The article "Inventory of good practices of sustainable and circular phosphorus management in the Visegrad Group (V4)" presents the activities in four EU member states in Eastern and Central Europe to manage phosphorus as a critical resource. The manuscript has significant lengths. For example, it does not contribute to the topic to present the position of phosphorus in the periodic table and explaining the four allotropes of elemental phosphorus. It would be more interesting to present the chemical identity of the natural deposits and the fertilizers derived from them and the extent to which they are bioavailable.
Answer: Thank you for your comments. The article deleted the section on the position of phosphorus in the periodic table and explaining the four allotropes of elemental phosphorus. Information on the chemical identity of the natural deposits and the fertilizers derived from them and the extent to which they are bioavailable was presented in articles: Smol, M. The importance of sustainable phosphorus management in the circular economy (CE) model: the Polish case study. J Mater Cycles Waste Manag 21, 227–238 (2019). https://doi.org/10.1007/s10163-018-0794-6 and Smol, M. The importance of sustainable phosphorus management in the circular economy (CE) model: the Polish case study. J Mater Cycles Waste Manag 21, 227–238 (2019). https://doi.org/10.1007/s10163-018-0794-6. For this reason, it was decided not to duplicate information on this topic in this article.
Reviewer comments: The reference should not be missing that the planetary boundary for phosphorus is clearly exceeded and the goal is to reduce the amount of emitted phosphate and to fertilze only accourding to the needs of the crops.
Answer: Thank you for your comments. Information has been added that the planetary limit of phosphorus has been clearly exceeded.
Reviewer comments: The goal to recycle phosphorus and to make accessible secondary phosphorus sources such as sewage sludge, biowaste and other wastes is clearly stated by the authors. Therefore, section 3.3 "Good practices of P recovery in V4 countries" is the key information. However, it is surprising that many manufacturers are listed without mentioning their P recovery activities.
Answer: Thank you for your detailed comments. Information related to the recovery of phosphorus in projects and activities mentioned in the article has been supplemented.
Reviewer comments: Only three projects are presented that focus on phosphorus recovery: PhosForce and PolFerAsh in Poland and the Slovak Grant Agency for Science.
Apparently, recovery of phosphorus from wastewater and sewage sludge does not take place in the V4 countries, even if mono-incineration is practised (page 10). An explanation of the economic reasons and a comparison with other EU countries should be added. The sustainable management of phosphorus also includes the development of agricultural practices that reduce the input of phosphorus into the environment and thus prevent damage such as the hypertrophication of Brno lake.
Answer: Thank you for your comment. Despite the current economic crisis, fertilizers from primary sources are still cheaper than fertilizers from secondary sources. For this reason, in the analysed countries, the topic of good practices in the context of obtaining alternative fertilizers from secondary sources is not very popular. Nevertheless, actions aimed at the development of this branch of the economy are being undertaken more and more often.
Thank you for your consideration. I hope you will accept the article for publication.
Prof. Dr. Marzena Smol

Reviewer 2 Report
The subject of the article is top important in the context of the need for resources as phosphorus used in many activities from agriculture to defense, in a circular economy of Europe as a whole and specific to Visegrad States as a particular case.
However, the authors in their research ,should be more explicit in the data they presented by underlying the links, with data, between internal production of phosphorus of the four countries, imported quantities, recycled sources, population growth, specific companies potential. All these data about production of phosphorus, imports of phosphorus, recycled phosphorus (obtained from ash, sludge etc.) and their relationship with the population number and investments in the projects of the companies with good practices, involved in P management are not clearly displayed, giving the sensation that is just a mention of their activities. A model (graphical, mathematical) to prove the links for all that data will be of interest.
The conclusions of the article are also too evasive and short.
Author Response
Dear Reviewer #2,
I am enclosing here with a manuscript entitled “Inventory of good practices of sustainable and circular phosphorus management in the Visegrad Group (V4)” submitted to “Resources” for possible evaluation after your review. With the submission of this manuscript I would like to undertake that the above mentioned manuscript has been corrected according to Your and other Reviewers comments.
Reviewer comments: The subject of the article is top important in the context of the need for resources as phosphorus used in many activities from agriculture to defense, in a circular economy of Europe as a whole and specific to Visegrad States as a particular case.
Answer: Thank you for your comments. Indeed the topic of the sustainable and circular phosphorus management in the Visegrad Group is important, primarily due to the fact that phosphorus is a critical raw material, and the region does not have its phosphorus deposits, therefore the demand for phosphorus raw materials is met only by imports.
Reviewer comments: The authors in their research, should be more explicit in the data they presented by underlying the links, with data, between internal production of phosphorus of the four countries, imported quantities, recycled sources, population growth, specific companies potential. All these data about production of phosphorus, imports of phosphorus, recycled phosphorus (obtained from ash, sludge etc.) and their relationship with the population number and investments in the projects of the companies with good practices, involved in P management are not clearly displayed, giving the sensation that is just a mention of their activities. A model (graphical, mathematical) to prove the links for all that data will be of interest.
Answer: Thank you very much for such an interesting solution. Unfortunately, due to the lack of data resulting from the lack of projects in the context of phosphate raw materials in the V4 countries, it is not possible to implement the proposed model. However, this will be the subject of further research in the PhosV4 project
Reviewer comments: The conclusions of the article are also too evasive and short.
Answer: The conclusions have been edited and supplemented.
Thank you for your consideration. I hope you will accept the article for publication.
Prof. Dr. Marzena Smol
